# Cryo-EM structure of the human somatostatin receptor 2 complex with its agonist somatostatin delineates the ligand-binding specificity

Yunseok Heo[1†], Eojin Yoon[2†], Ye-Eun Jeon[1], Ji-Hye Yun[1,3], Naito Ishimoto[4], Hyeonuk Woo[5], Sam-Yong Park[4], Ji-Joon Song[2,3]*, Weontae Lee[1,3]*

[1]Department of Biochemistry, College of Life Science and Biotechnology, Yonsei University, Seoul, Republic of Korea; [2]Department of Biological Sciences, Korea Advanced Institute of Science and Technology, Daejeon, Republic of Korea; [3]PCG-Biotech, Ltd., 508 KBIZ DMC Tower, Sangam-Ro, Seoul, Republic of Korea; [4]Drug Design Laboratory, Graduate School of Medical Life Science, Yokohama City University, Yokohama, Japan; [5]Department of Chemistry, Seoul National University, Seoul, Republic of Korea

**\*For correspondence:**
songj@kaist.ac.kr (J-JoonS);
wtlee@yonsei.ac.kr (WL)

†These authors contributed equally to this work

**Abstract** Somatostatin is a peptide hormone that regulates endocrine systems by binding to G-protein-coupled somatostatin receptors. Somatostatin receptor 2 (SSTR2) is a human somatostatin receptor and is highly implicated in hormone disorders, cancers, and neurological diseases. Here, we report the high-resolution cryo-EM structure of full-length human SSTR2 bound to the agonist somatostatin (SST-14) in complex with inhibitory G (G$_i$) proteins. Our structural and mutagenesis analyses show that seven transmembrane helices form a deep pocket for ligand binding and that SSTR2 recognizes the highly conserved Trp-Lys motif of SST-14 at the bottom of the pocket. Furthermore, our sequence analysis combined with AlphaFold modeled structures of other SSTR isoforms provide a structural basis for the mechanism by which SSTR family proteins specifically interact with their cognate ligands. This work provides the first glimpse into the molecular recognition mechanism of somatostatin receptors and a crucial resource to develop therapeutics targeting somatostatin receptors.

## Editor's evaluation

This manuscript reports the cryoEM structure of somatostatin receptor 2 (SSTR2) bound to its agonist SST– 14 and a heterotrimeric G protein. In addition to presenting the structure itself, the authors include discussion and analysis of ligand recognition and subtype specificity, guided by AlphaFold2 modeling of other somatostatin receptor subtypes. Site– directed mutagenesis and signaling assay data attest to the importance of receptor– ligand contacts that contribute to subtype specificity. Somatostatin signaling is important in endocrine biology, including in diseases such as acromegaly and some cancers, and structures of somatostatin receptors will help illuminate the molecular details of somatostatin receptor signal transduction. Of note, the structure of SSTR2 was also separately reported by Robertson et al., Nat. Struct. Mol. Biol. 2022 while this manuscript was under review.

## Introduction

Somatostatin (SST) is a cyclic peptide hormone that regulates neurotransmission and hormone secretion (*Weckbecker et al., 2003*; *Saito et al., 2005*; *Morrison et al., 1985*). SST was initially identified as an inhibitory hormone produced in hypothalamic neurons, but it is also released in the gastrointestinal

(GI) tract. SST binds to G-protein-coupled somatostatin receptors (SSTRs), inhibiting adenylyl cyclase via inhibitory G-proteins (*Patel et al., 1994*; *Patel et al., 1995*). SSTRs are class A G-protein-coupled receptors (GPCRs), and there are five isoforms (SSTR1–SSTR5) (*Leu and Nandi, 2010*). Each SSTR isoform is expressed in different tissues and organs and has a distinct function (*Bartha and Győrffy, 2021*). SST inhibits the secretion of growth hormone from the pituitary gland, and SST secreted in the GI tract inhibits the secretion of GI hormones. Furthermore, SST in the brain has been shown to modulate cortical circuits (*Song et al., 2020*). Therefore, the SST-SSTR axis is highly implicated in several human diseases, including acromegaly, cancers, and neurological disorders (*Song et al., 2020*; *Song et al., 2021*; *Lamberts et al., 2002*). Several SST analogs, such as octreotide, lanreotide, and pasireotide, targeting SSTRs have been developed, and they are already in clinical use (*Lamberts et al., 2002*). For example, acromegaly is caused by excessive production of growth hormone in the pituitary gland, and SST analogs such as octretotide and lanreotide are clinically used for treating acromegaly by exploiting the inhibitory role of SST via SSTR. However, the currently used SST and its analogs have obscure binding specificity among SSTR isoforms, making it difficult to develop means to modulate each isoform specifically while minimizing off-target effects. SSTR2 is mainly expressed in brain and endocrine tissues and is implicated in neuroendocrine tumors and Alzheimer's disease (*Song et al., 2021*; *Uhlén et al., 2015*). To understand the molecular and structural basis of the SSTR interaction with its ligand, we determined the cryo-EM structure of SSTR2 bound to its endogenous agonist somatostatin (SST-14) in a complex with inhibitory G-proteins. Our structural work provides insight into the mechanism by which SSTRs recognize their ligands and will serve as a platform to develop selective agonists and therapeutics.

## Results and discussion

To investigate the molecular basis of ligand-specific binding by SSTR2, we determined the cryo-EM structure of human SSTR2 bound to the SST-14 somatostatin peptide. Full-length human SSTR2 in a thermostabilized apocytochrome b562 RIL (BRIL) fusion form and a heterotrimeric Gαi1/Gβ1γ2 complex with Gαi1-recognizing scFv16 were separately expressed and purified, and the SSTR2-Gαi1/Gβ1γ2-scFv16 complex in the presence of SST-14 cyclic peptide was prepared for structural study (*Figure 1—figure supplement 1*). The complex was plunge-frozen, and micrographs were collected using a Titan Krios 300 keV with a Gatan K2 Summit direct detector in movie mode. The collected data were processed and refined with Relion 3.1 (*Zivanov et al., 2018*; *Figure 1—figure supplement 2* and *Table 1*). During refinement, SSTR2 was separated into two bodies (Body 1: SSTR2 + Gαi1β1 and Body 2: Gβ1γ2 + scFv16), and the two bodies were separately refined. The overall resolution of the whole SSTR2-Gαi1/Gβ1γ2-scFv16 complex was estimated as 3.72 Å at 0.143 criteria of the gold standard FSC, and the resolutions of Body 1 and Body 2 were estimated as 3.65 and 3.22 Å, respectively. The cryo-EM map was well resolved, and the atomic model for most part of the SSTR2 complex was built on the map (*Figure 1A* and *Figure 1—figure supplement 3*).

In the structure, the C-terminal helix of Gαi1 is inserted inside of the seven transmembrane (TM) helices forming hydrophobic interactions with TM 5–7 helices (*Figure 1B*), and scFv16 interacts with the N-terminal helix of Gαi1 and a loop protruding from Gβ1, stabilizing the structures. The topology and structure of the TM helices is similar to other known GPCRs (*Weis and Kobilka, 2018*). We compared our active SSTR2 structure with the inactive structures of oxytocin receptor (OTR), Alpha-Fold predicted SSTR2, and the active structure of melanocortin receptor 4 (MC4R) (*Waltenspühl et al., 2020*; *Israeli et al., 2021*; *Jumper et al., 2021*). Our cryo-EM structure of SSTR2 shows a conventional active conformation, in which TM5, TM6, and TM7 are displaced outward from TM3 (*Figure 1A* and *Figure 1—figure supplement 4*).

The TM helices of SSTR2 at the extracellular side form a pocket for ligands, and SST-14 nestles snugly at the pocket (*Figure 1C*). Among the 14 amino acids in SST-14, 12 amino acid residues from Cys3 to Cys14 are clearly visible in the cryo-EM map, and the positions of the side chains were unambiguously assigned on the map (*Figure 2A*). SST-14 is cyclized via a disulfide bond between Cys3 and Cys14, and the region between Phe6 and Phe11 forms a flat sheet (*Figure 2B*). In this configuration, the disulfide bond is located outward from the ligand-binding pocket of SSTR2, and the two amino acids of Trp8 and Lys9 are positioned at the bottom of the binding pocket (*Figure 2B*). The binding mode of SST-14 to SSTR2 is similar to those of other cyclic peptides, including oxytocin in OTR and setmelanotide in MC4R (*Israeli et al., 2021*; *Meyerowitz et al., 2022*; *Figure 1—figure supplement*

**Table 1.** Refinement statistics.

| Data collection and processing | | |
|---|---|---|
| **Magnification** | | |
| Voltage (kV) | | 300 |
| Total electron exposure/used (e/Å$^2$) | | 55.04/29.24 |
| Defocus range (μm) | | −0.8 ~ −2.0 |
| Pixel size (Å$^2$) | | 0.829 |
| Processing program | | Relion 3.1 |
| Obtained/used micrographs (no.) | | 5523/5523 |
| Initial/final particles used (no.) | | 6,677,042/320,885 |
| Symmetry imposed | | C1 |
| Resolution (Å) (FSC threshold) | | |
| GPCR + G-protein | | 3.72 (0.143) |
| | GPCR + G$_{αβ}$ | 3.65 (0.143) |
| Multibody refinement | G$_{βγ}$+scfv16 | 3.22 (0.143) |
| **Refinement** | | |
| Refinement program | | PHENIX |
| Model composition | | |
| Nonhydrogen atoms | | 8660 |
| Protein residues | | 1129 |
| r.m.s. deviation | | |
| Bond length (Å) | | 0.004 |
| Bond angle (°) | | 0.547 |
| Validation | | |
| MolProbity score | | 2.08 |
| Clash score | | 7.79 |
| Ramachandran plot (%) | | |
| Favored/allowed/outliers | | 95.86/4.24/0.0 |
| Mask CC | | 0.80 |

*5*). The disulfide bonds of the cyclic peptides are located at the extracellular surface of the ligand-binding pockets formed by TMs. While SSTR2 recognizes the Trp-Lys motif in SST-14, OTR and MC4R interact with the Tyr-Ile motifs in oxytocin and dPhe-Arg-Trp in setmelanotide via the bottom of the binding pockets, respectively.

To understand the specific recognition of SST-14 by SSTR2, we examined the detailed interactions between SST-14 and SSTR2. At the bottom of the ligand-binding pocket, Trp9 of SST-14 interacts with SSTR2 via a hydrophobic pocket formed by Ile177, Phe208, Thr212, and Phe272 from TM4–6 helices (*Figure 2C*). Notably, Lys9 is the only charged residue among the amino acids of SST-14, which is located inside of the pocket. Lys9 of SST-14 forms a salt bridge with Asp122 with a 2.7 Å distance and makes a hydrogen bond with the oxygen atom of the Gln126 side chain. In addition, the carbonyl oxygen in the peptide bond between Lys8 and Trp9 forms a hydrogen bond with Asn276. While the region of S$_{13}$C$_{14}$C$_3$K$_4$ is exposed to the solvent, other hydrophobic residues form stable interactions with the hydrophobic residues in the ligand-binding pocket (*Figure 2C*). Specifically, the Phe275, Leu290, and Phe294 residues of SSTR2 accommodate the Phe6 residue of SST-14, and the Phe7 residue of SST-14 is stacked on Tyr205 coming from the TM5 helix and further stabilized by the interactions

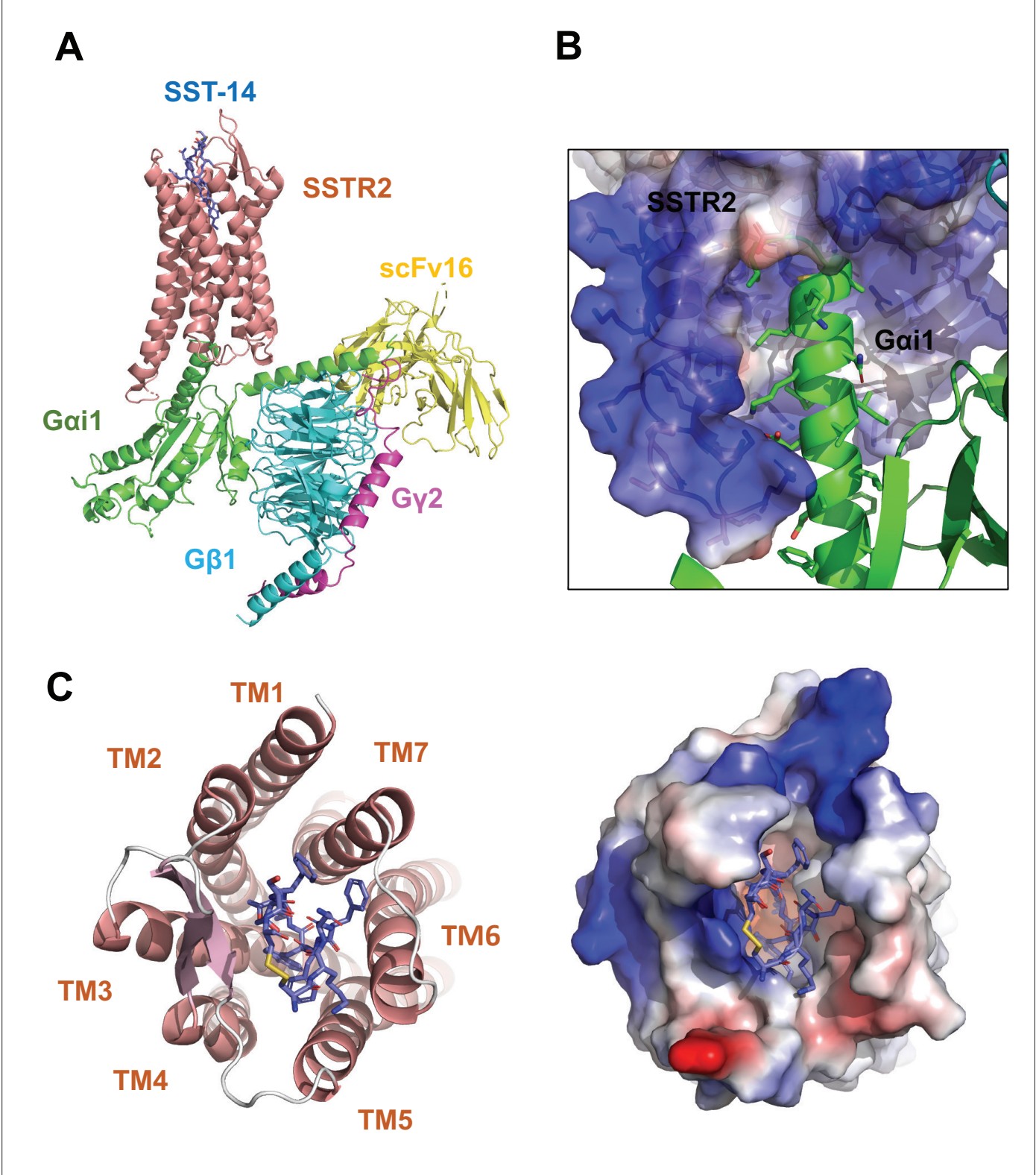

**Figure 1.** Cryo-EM structure of somatostatin receptor 2 (SSTR2) G-protein complex with SST-14. (**A**) The atomic model of the SSTR2 complex is shown in a ribbon model. SSTR2, Gαi1, Gβ1, Gγ2, and scFv16 are shown in a ribbon diagram and colored in salmon, green, cyan, magenta, and yellow, respectively. The bound SST-14 cyclic peptide is shown in a ball-and-stick model and colored in navy blue. (**B**) The C-terminal of Gαi1 is inserted in the

*Figure 1 continued on next page*

*Figure 1 continued*

pocket formed by TM5–7 of SSTR2 forming hydrophobic interactions. (**C**) SST-14 (shown in a ball-and-stick) is bound to the pocket formed by seven TMs of SSTR2 (colored in salmon) at the extracellular side (left panel). SSTR2 is shown in an electrostatic surface representation (right panel).

The online version of this article includes the following source data and figure supplement(s) for figure 1:

**Figure supplement 1.** Purification of recombinant human somatostatin receptor 2 (SSTR2)-Gαi1/Gβ1γ2-scFv16 complex.

**Figure supplement 1—source data 1.** An uncropped gel for the purification.

**Figure supplement 2.** Cryo-EM processing.

**Figure supplement 3.** Cryo-EM structure of somatostatin receptor 2 (SSTR2) complex.

**Figure supplement 4.** Comparison among somatostatin receptor 2 (SSTR2) and other G-protein-coupled receptor (GPCR) structures.

**Figure supplement 5.** The binding modes of cyclic peptide ligands of somatostatin receptor 2 (SSTR2), oxytocin receptor (OTR), and melanocortin receptor 4 (MC4R) G-protein-coupled receptors (GPCRs).

with the Ile195 and Phe208 residues. To validate the recognition of SST-14 by SSTR2 observed in the cryo-EM structure, we mutagenized several critical residues involved in the interaction and performed a functional assay in HEK293 cells. We measured the degree of inhibition of cAMP generation upon forskolin stimulation by homogeneous time-resolved fluorescence resonance energy transfer (HTRF-FRET) (*Figure 2D* and *Figure 2—figure supplement 1*). We first eliminated the salt bridge between Lys9 of SST-14 and SSTR2 by mutating Asp122 to alanine. While the cAMP production was completely inhibited by SST-14 at submicromolar range concentrations for SSTR2$_{WT}$, SSTR2$_{D122A}$ showed less than 20% inhibition of cAMP production, indicating that the disruption of the bridge likely abrogated the interaction between SSTR2 and SST-14. Next, we mutated Gln126 to methionine and examined its effect. Gln126 forms a hydrogen bond with Lys9 of SST-14 in addition to the salt bridge, and mutating Gln to Met substantially decreased the function of SSTR2. Notably, among the five isoforms, SSTR2, SSTR3, and SSTR5 isoforms have glutamine while SSTR1 and SSTR4 have methionines at this position. We further examined the effects of the hydrophobic residues interacting with SST-14 by mutating Phe272 or Phe294 to alanines. Eliminating the hydrophobic interactions between SST-14 and SSTR2 also substantially decreased the function of SSTR2, emphasizing their roles in recognizing its ligand. Combined with the cryo-EM structure, our functional analysis further delineates the specific recognition of SSTR2 for its ligand.

The endogenous agonists for SSTRs are SST-14 and SST-28 (*Patel et al., 1994*; *Patel and Srikant, 1994*). Compared to SST-14, SST-28 has an extra 14 residues at the N-terminus (*Figure 3A*). Our cryo-EM structure shows that the region of SST-14 is likely sufficient for binding to SSTR2. Consistent with this hypothesis, SST-28 has a binding affinity similar to that of SST-14, in the subnanomolar range (*Song et al., 2021*). In addition to SST-14, there are several other ligands and drugs that bind to SSTR2, including cortistatin, octreotide, pasireotide, and lanreotide. All of these ligands and drugs are cyclic forms of peptides and contain absolutely conserved Trp and Lys residues (*Figure 3A*). The interaction analysis based on the cryo-EM structure showed that these highly conserved Trp and Lys residues tightly interact with SSTR2 via hydrophobic interactions and a salt bridge. Therefore, it is likely that the Trp-Lys motif of the ligands and drugs is a major determinant for binding.

There are five isoforms of SSTRs (SSTR1–5), which are classified into two families based on their structural and pharmacological properties. SSTR2, SSTR3, and SSTR5 belong to the SRIF1 receptor family, and SSTR1 and SSTR4 belong to the SRIF2 receptor family (*Günther et al., 2018*). The tissues that express each isoform and its cognate ligand differ, implying the isoform-specific function of SSTRs (*Song et al., 2021*). To further investigate the ligand specificity of the isoforms, we further examined the ligand-binding pocket of SSTRs by combining a sequence alignment analysis, our cryo-EM structure of SSTR2 and predicted models of other isoforms from the AlphaFold database (https://alphafold.ebi.ac.uk) (*Jumper et al., 2021*). To examine the sequence conservation of the ligand-binding pocket, we colored the residues near the ligand-binding pocket of SSTR2 in red or yellow depending on the degree of conservation based on a sequence alignment among SSTRs (*Figure 3B and C*). This practice revealed that the residues involved in interacting with the Trp-Lys motif are highly conserved. On the other hand, the residues interacting with the other part of SST-14 are highly variable, suggesting that this region contributes to the ligand specificity among SSTR isoforms.

To further understand the binding specificity of SSTR isoforms, we superimposed modeled structures of SSTR1, SSTR3, SSTR4, and SSTR5 isoforms from the AlphaFold Structure Database (alphafold.

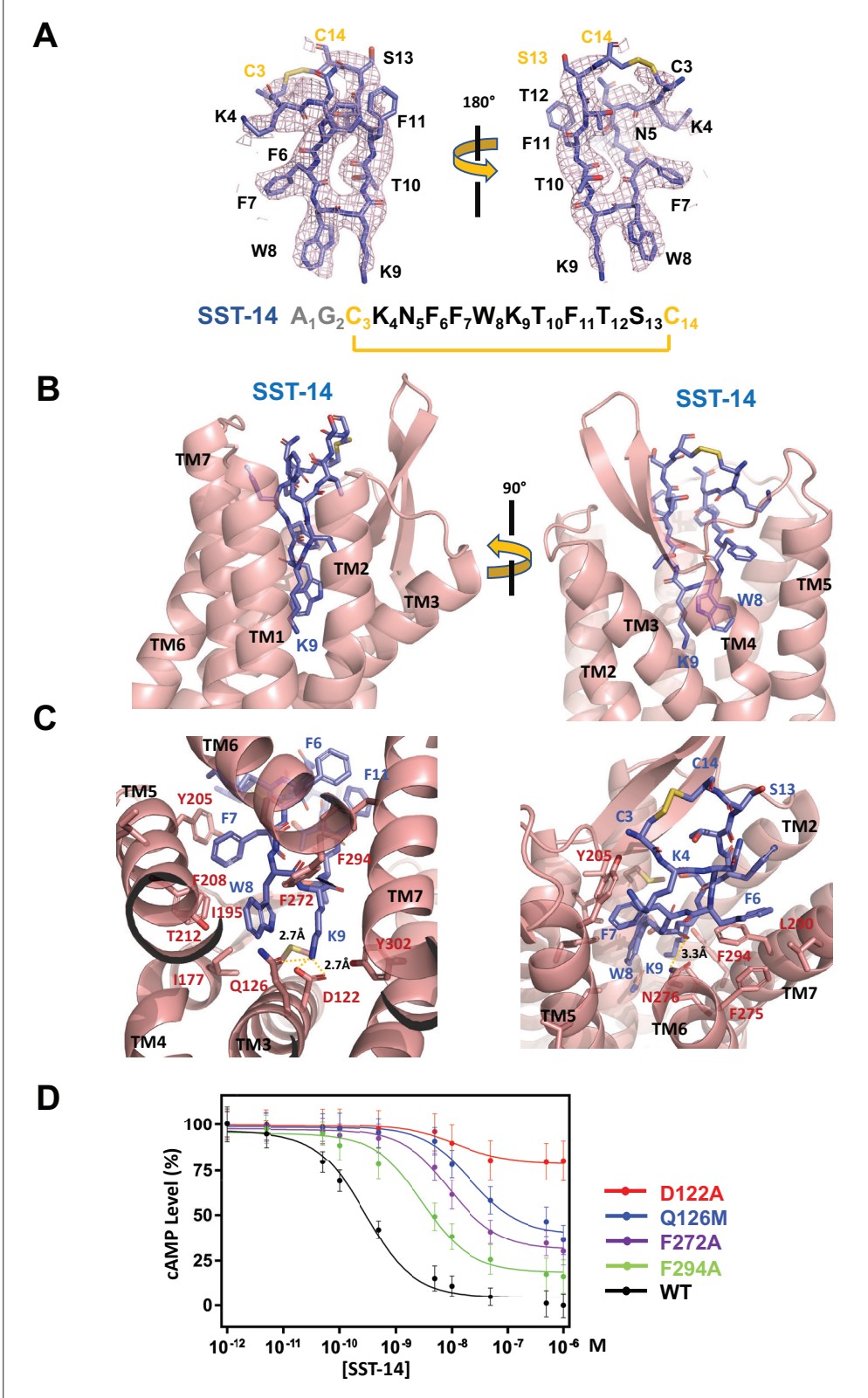

**Figure 2.** Somatostatin receptor 2 (SSTR2) recognition of SST-14 ligand. (**A**) The cryo-EM map near the SST-14 ligand in two different orientations. The sequence of SST-14 is drawn at the below. Two amino acids, which are not visible, are colored in gray. $C_3$ and $C_{14}$ make a covalent bond to form a cyclic peptide. (**B**) The ligand-binding pocket of SSTR2 with SST-14 is shown in two different orientations. SSTR2 is shown in a ribbon diagram and SST-14

*Figure 2 continued on next page*

*Figure 2 continued*

in a stick model. (**C**) Detailed interactions between SST-14 and SSTR2. The salt bridge between Lys9 and Asp122, and a hydrogen bonding between Lys9 and Gln126 are indicated with yellow dotted lines. (**D**) The dose-signal curves of SSTR2 for SST-14. WT, D122A, Q126M, F272A, F294A of SSTR2, measuring the inhibition of cAMP production upon forskolin stimulation (n = 3).

The online version of this article includes the following source data and figure supplement(s) for figure 2:

**Figure supplement 1.** Western blot analysis of somatostatin receptor (SSTR) expression in HEK293 cells.

**Figure supplement 1—source data 1.** Uncropped Western blot images.

ebi.ac.uk) (*Jumper et al., 2021*) on the cryo-EM structure of SSTR2 (*Figure 3—figure supplement 1*) and examined the interactions between SST-14 bound to SSTRs. As the AlphaFold models were generated in the absence of ligands, the modeled structures represent an inactive state where the ligand is not bound. Among the residues in the ligand-binding pockets, residues at three positions, which directly interact with SST-14, vary among the isoforms (*Figure 3D*). The first position is at Gln126 in SSTR2 located in the bottom of the ligand-binding pocket and involved in interacting with the Trp-Lys motif of SST-14. SSTR3 and SSTR5 also have Gln at this position while SSTR1 and SSTR4 have Mets. As this position is located between Trp8 and Lys9 of SST-14, Gln interacts with Lys9, while Met interacts with Trp8. Interestingly, our functional assay with Gln126Met mutant showed a substantial decrease in the function of SSTR2, suggesting that Gln126 may play a role in the substrate specificity of the isoforms. The second position is Tyr205, which interacts with Phe7 of SST-14 via stacking interactions between aromatic rings. Each isoform has a different residue at this position (Leu220 in SSTR1, Arg203 in SSTR3, Ser208 in SSTR4, and Gly198 in SSTR5). Therefore, it is possible that the residues at this position contribute to the binding specificity of each isoform. The third position is at Phe294, which is located on TM7 in SSTR2. Phe294 holds Phe6 of SST-14 via hydrophobic interactions. While SSTR3 and SSTR5 have Tyr residues at this position, maintaining the hydrophobicity, SSTR1 and SSTR4 have Ser305 and Asn293 residues at this position, respectively. Collectively, our structural and sequence analyses suggest that SSTR2, SSTR3, and SSTR5 likely have different binding characteristics than SSTR1 and SSTR4. Consistent with this hypothesis, octreotide, lanreotide, and pasireotide have higher affinities toward SSTR2, SSTR3, and SSTR5 than SSTR1 and SSTR4 (*Song et al., 2021*; *Barbieri et al., 2013*). A recent report on the SSTR2 structure by Robertson et al. showed that extracellular loop 2 (ECL2) and ECL3 are involved in ligand-specific binding among SSTR isoforms, which have highly variable sequences among the SSTR isoforms (*Figure 3B*; *Robertson et al., 2022*). In addition, consistent with our findings, they also showed that the interactions between the ligands and SSTR2 at the inside of the binding pocket are also critical for the ligand specificity. Therefore, several regions of SSTR2 including residues in the ECLs and the inside of the binding pockets likely contribute to the ligand-binding specificity. Together with the recent work, our structural and sequence analysis revealed subtle differences in the ligand-binding pocket in each SSTR isoform, providing a critical information to understand how each SSTR isoform specifically recognizes its cognate ligand and drug.

In conclusion, our work on the structure of the SSTR2 and SST-14 ligand complex delineates the specific ligand recognition by SSTRs. Furthermore, as SSTRs are highly implicated in several human diseases, our works on SSTR2 and the SST-14 complex will serve as a fundamental platform to design novel and specific therapeutics to modulate SSTRs.

# Materials and methods
## Construct design
WT full-length human SSTR2 inserted with an N-terminal hemagglutinin signal sequence and FLAG tag inserted, followed by an 8× His tag and HRV 3C protease cleavage site was cloned into the modified pFastBac1 (Invitrogen, Carlsbad, CA) vector. For the SSTR2 stability, an additional sequence of A1-L106 encoding thermostabilized apocytochrome BRIL with mutations (M7W, H102I, R106L) was added after the 8× His tag at the N-terminus (*Chun et al., 2012*). The heterotrimeric Gαi1/Gβ1γ2 was designed as previously described (*Kim et al., 2020*). The scFv16 single-chain antibody containing the GP67 secretion signal sequence was also inserted into the pFastBac1 vector (*Maeda et al., 2018*).

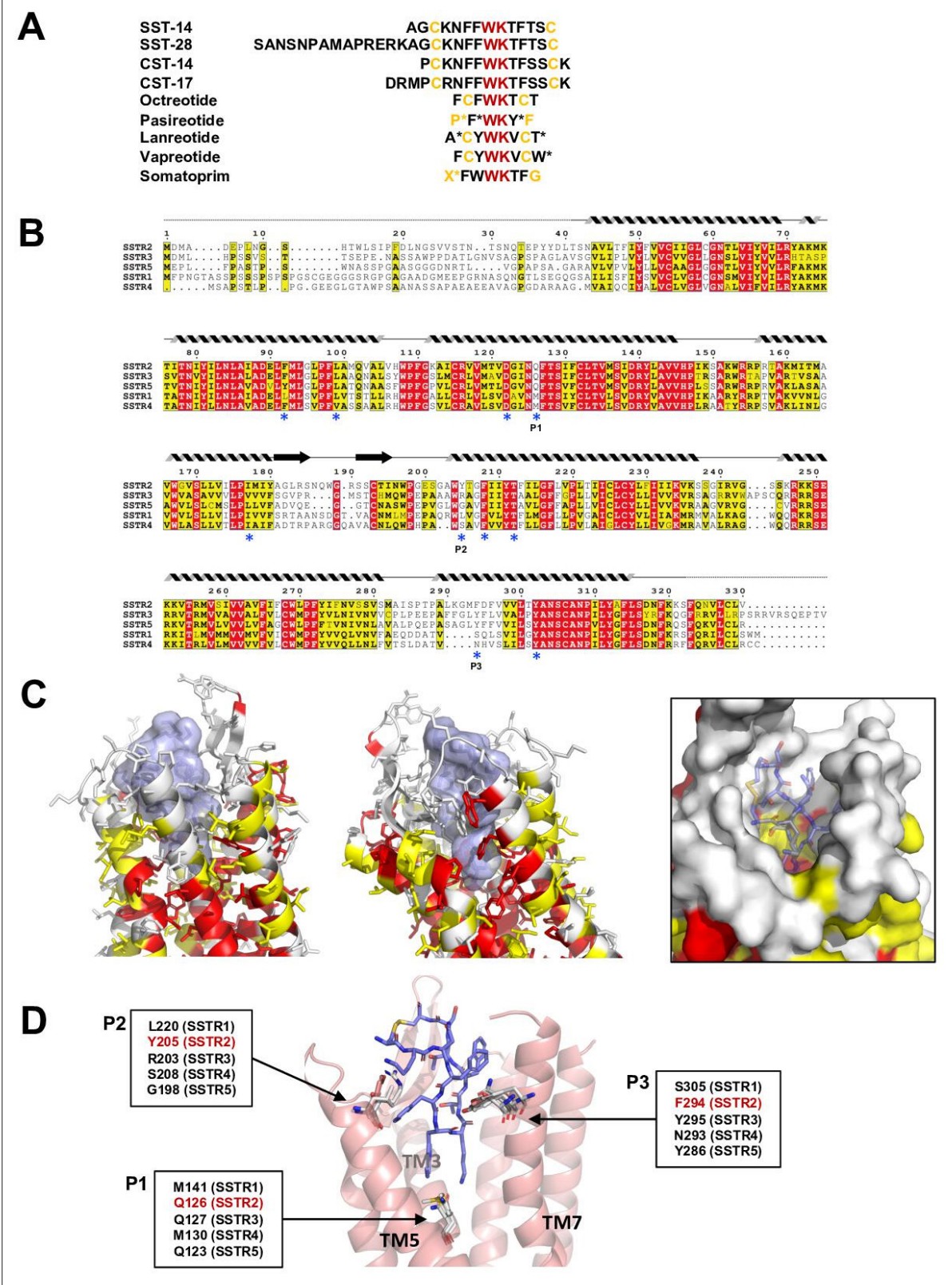

**Figure 3.** Sequence and structural analysis of ligand-binding pockets of somatostatin receptor (SSTR) isoforms. (**A**) The sequences of somatostatins and its analogs. Trp-Lys motif is absolutely conserved among all SSTR-binding ligands, emphasizing the importance of the Trp-Lys motif for the function. Modified amino acids are marked with asterisks in pasireotide (P*: hydroxyproline, F*: 2-phenylglycine, Y*: phenylmethylated tyrosine), lanreotide (A*: D-2-naphthylalanine, T*: amidated threonine), vapreotide (W*: amidated tryptophan), and somatoprim (X*: ω-amino acid). (**B**) A sequence alignment

*Figure 3 continued on next page*

*Figure 3 continued*

among SSTR isoforms. The amino acids are colored in red or yellow depending the degree of sequence conservation among the isoforms. The secondary structures based on the cryo-EM structure of SSTR2 are shown above the sequences. Critical residues recognizing SST-14 are indicated with asterisks under the sequences. P1, P2, and P3 indicate highly variable sequences among the isoforms. (C) The conserved amino acids near the ligand-binding pocket are colored according to the sequence alignment in (B). Highly conserved amino acids are clustered near the inner bottom of the pocket, which interacts with the Trp-Lys motif, while amino acids near the upper part of the pocket are not well conserved. (D) Three variable positions (P1, P2, P3) in the ligand-binding pockets based on the cryo-EM structure of SSTR2 and AlphaFold modeled structure of SSTR1, SSTR3, SSTR4, and SSTR5.

The online version of this article includes the following figure supplement(s) for figure 3:

**Figure supplement 1.** AlphaFold modeled structures of somatostatin receptor (SSTR) isoforms.

## Expression of SSTR2
### The Gαi/Gβ γ heterotrimer and scFv16

SSTR2 and Gαi1/Gβ1γ2 were expressed using the Bac-to-Bac Baculovirus Expression system (Invitrogen) in *Spodoptera frugiperda* (Sf9) cells using ESF media (Expression Systems, Davis, CA). Using the high titer virus at a multiplicity of infection of 3, Sf9 cells at a density of $2 \times 10^6$ cells/mL in 400 mL of biomass were infected. The cells were incubated with shaking at 27°C for 72 hr and harvested, washed with phosphate-buffered saline (PBS), flash-frozen in liquid nitrogen, and stored at −80°C until further use. The scFV16 single-chain antibody was expressed using the Bac-to-Bac expression system in Sf9 cells, and a high titer virus was generated. The cells were incubated with shaking at 27°C for 72 hr, and secreted scFv16 in the supernatant was separated from the cells by centrifugation.

## Purification of SSTR2
### The Gαi/Gβ γ heterotrimer and scFv16

SSTR2 frozen pellets were thawed and resuspended at 4°C with the addition of EDTA-free protease inhibitor cocktail (Sigma Aldrich, St Louis, MO). The cell membranes were obtained by repeated lysis and Dounce homogenization using hypotonic buffer containing 10 mM HEPES (pH 7.5), 10 mM MgCl$_2$, 20 mM KCl, and protease inhibitors and hypertonic buffer containing 10 mM HEPES (pH 7.5), 10 mM MgCl$_2$, 20 mM KCl, 1.0 M NaCl, and protease inhibitors. The washed membrane fractions were resuspended in buffer containing 30 mM HEPES (pH 7.5), 5 mM MgCl$_2$, 10 mM KCl, 500 mM NaCl, 200 µM SST-14, and protease inhibitors. Then, membrane fractions were incubated at 25°C for 1 hr and solubilized in 1% (w/v) n-dodecyl-β-D-maltopyranoside (DDM) (Anatrace, Maumee, OH) and 0.2% (w/v) cholesteryl hemisuccinate (CHS) (Anatrace, Maumee, OH) at 4°C for 3 hr. The solubilized solution was isolated by ultracentrifugation at 150,000 × g for 60 min, and the supernatant was isolated. TALON IMAC (Clontech) resin was added to the supernatant. The mixture was incubated at 4°C overnight. After incubation, the resin-bound SSTR2 was loaded onto a disposable chromatography column (Bio-Rad, Hercules, CA), and the resin was washed with 20 column volumes (CVs) of wash buffer containing 50 mM HEPES (pH 7.5), 500 mM NaCl, 10 mM MgCl$_2$, 1% (w/v) DDM, 0.2% CHS (w/v), 5 mM imidazole, 10% (v/v) glycerol, and 50 µM SST-14. Bound proteins were eluted with 10 CVs of elution buffer containing 50 mM HEPES (pH 7.5), 500 mM NaCl, 0.05% (w/v) lauryl maltose neopentyl glycol (LMNG) (Anatrace, Maumee, OH), 0.005% (w/v) CHS, 300 mM imidazole, 10% (v/v) glycerol, and 100 µM SST-14. PD-10 desalting column (Cytiva) was used to remove the high concentration of imidazole. SSTR2 was then treated overnight at 4°C with HRV 3C protease. A reverse affinity column was used for the further purification of untagged SSTR2 with buffer containing 50 mM HEPES (pH 7.5), 500 mM NaCl, 0.05% (w/v) LMNG, 0.005% (w/v) CHS, 10% (v/v) glycerol, and 50 µM SST-14. SSTR2 was collected and concentrated and loaded onto a Superdex 200 Increase 10/300 GL column (Cytiva) with buffer containing 20 mM HEPES (pH 7.5), 100 mM NaCl, 1 mM MgCl$_2$, 0.5 mM TCEP, 0.05% (w/v) LMNG, 0.005% (w/v) CHS, and 50 µM SST-14 via ÄKTA pure system (Cytiva). Fresh SSTR2 was used for SSTR2-Gαi1/Gβ1γ2 complex formation. Gαi1/Gβ1γ2 frozen pellets were thawed and resuspended at 4°C with the addition of a protease inhibitor cocktail. The cells were lysed in lysis buffer containing 20 mM HEPES (pH 7.5), 100 mM NaCl, 1 mM MgCl$_2$, 20 mM imidazole, 5 mM β-mercaptoethanol, 100 µM GDP, 1% (v/v) Tergitol-type NP-40 (Sigma), and protease inhibitors. The soluble fraction was isolated by ultracentrifugation at 130,000 × g at 4°C for 30 min. The Gi heterotrimer in the soluble fraction was purified using Ni-NTA chromatography and eluted with buffer containing

20 mM HEPES (pH 7.5), 100 mM NaCl, 1 mM MgCl$_2$, 300 mM imidazole, 5 mM β-mercaptoethanol, and 10 µM GDP. HRV 3C protease was added and the 6× His tag was cleaved at 4°C overnight. A reverse affinity column was used for purification of untagged Gαi1/Gβ1γ2. The untagged Gαi1/Gβ1γ2 protein was further purified by size exclusion chromatography (SEC) on a HiLoad 16/600 Superdex 200 column (Cytiva) with the following buffer: 20 mM HEPES (pH 7.5), 100 mM NaCl, 1 mM MgCl$_2$, 500 µM TCEP, and 10 µM GDP. The eluted protein was concentrated to 5 mg/mL and stored at −80°C until further use. The supernatant containing scFv16 was loaded onto a HisTrap EXCEL column. The column was washed with 10 CVs of wash buffer containing 20 mM HEPES (pH 7.5), 100 mM NaCl, and 50 mM imidazole. The bound protein was eluted using the same buffer supplemented with 500 mM imidazole. After the eluted protein was concentrated, PD-10 desalting column was used to remove the high concentration of imidazole. The C-terminal 6× His tag was cleaved by incubation with HRV 3C protease at 4°C overnight. A reverse affinity column was used for purification of untagged scFv16. scFv16 was further purified by SEC on a HiLoad 16/600 Superdex 200 column (Cytiva) with following buffer containing 20 mM HEPES (pH 7.5) and 100 mM NaCl. Monomeric fractions were pooled, concentrated, and flash-frozen in liquid nitrogen until further use.

## Formation of SSTR2-Gαi1/Gβ1γ2 complex

To form the SSTR2-Gαi1/Gβ1γ2-scFv16 complex, fresh SST-14-bound SSTR2 was mixed with a 1.2 molar excess of Gαi1/Gβ1γ2. The coupling reaction was performed at 25°C for 1 hr, followed by the addition of 0.2 units/mL apyrase (New England Biolabs). After an additional 1 hr at 25°C, lambda phosphatase (New England Biolabs) was added. To form the SSTR2-Gαi1/Gβ1γ2-scFv16 complex, a 1.2 molar excess of scFv16 was added to the SSTR2-Gαi1/Gβ1γ2 complex and further incubated at 4°C overnight. The SSTR2-Gαi1/Gβ1γ2-scFv16 complex sample in LMNG/CHS-containing buffer was loaded on a Superdex 200 Increase 10/300 GL column (Cytiva) equilibrated in buffer containing 20 mM HEPES (pH 7.5), 100 mM NaCl, 1 mM MgCl$_2$, 0.5 mM TCEP, 0.001% (w/v) LMNG, 0.0001% (w/v) CHS, and 40 µM SST-14. Peak fractions were concentrated to 2.5 mg/mL for electron microscopy studies.

## Cryo-EM image processing

Cryo-EM data were collected with a Titan Krios, Yokohama, Japan. A total of 5523 movies were collected in electron counting mode for 50 frames with a total dose of 55.04 e/Å (*Saito et al., 2005*). Magnification of micrographs was ×75,000, 0.867 Å/pixel. After data collection, image processing was performed by Relion 3.1 (*Zivanov et al., 2018*) in the SBGrid package (https://www.sbgrid.org/; *Morin et al., 2013*). Initially, collected movies were motion-corrected by MotionCorr2 and ctffind using CTFFIND 4.1 embedded in Relion 3.1. Then, 6,677,042 particles were picked by template-based autopicking. Bad particles were filtered out through several rounds of 2D classification until secondary structures were visible in 2D classes. After 2D classification, the selected 2,906,685 particles were subjected to 3D classification with C1 symmetry dividing the particles into eight classes. Among eight 3D classes, three high-resolution classes were selected. Several rounds of 3D classification were performed until the final resolution reached 4.08 Å by 3D autorefining with a final of 320,885 particles. The final particles were reextracted from motion-corrected micrographs with a total dose of 29.24 e$^-$/Å (*Saito et al., 2005*), and the resolution was improved up to 3.72 Å (FSC threshold 0.143). For further processing, the whole model was divided into two bodies for multibody refinement. Body 1 contains SSTR2 + Gαi1/Gβ1 + scFv16, and Body 2 contains Gβ1/Gγ2 + scFv16. The resolution after multibody refinement and sharpening was 3.65 Å for Body 1 and 3.22 Å for Body 2 (FSC threshold 0.143). The atomic model was built on the cryo-EM map with the help of the available Gαi1/Gβ1 + scFv16 structure and the AlphaFold predicted SSTR2 structure. The cryo-EM map and the model were deposited at the EMDB (https://www.ebi.ac.uk/) and RCSB (https://www.rcsb.org/) databases with the accession codes of EMD-32543 and 7WJ5, respectively.

## cAMP functional assay

Inhibition of forskolin-stimulated cAMP production was measured using a CISBIO cAMP HTRF-FRET kit (Cisbio, Bedford, MA). First, 2 µg of SSTR2 DNA was transfected into HEK293 cells, and the cells were plated in 96-well plates (4 × 10$^4$/well) in 5 µL using DMEM supplemented with 1% FBS following suspension in 5 mM EDTA/PBS. The cells were treated with 2 µL of 50 µM phosphodiesterase inhibitor

(RO-20–1724) dissolved in a stimulation buffer (1× DPBS containing 0.1% BSA) and 2 µL of SST-14 at 10 different concentrations ($5 \times 10^{-6}$, $2.5 \times 10^{-6}$, $2.5 \times 10^{-7}$, $5 \times 10^{-8}$, $2.5 \times 10^{-8}$, $2.5 \times 10^{-9}$, $5 \times 10^{-10}$, $2.5 \times 10^{-10}$, $2.5 \times 10^{-11}$, and $5 \times 10^{-12}$ M) dissolved in distilled water, and 1 µL of 1 µM forskolin for 45 min at 37°C. The plates were developed by adding 5 µL of anti-cAMP cryptate and 5 µL of d2-labeled cAMP in a lysis buffer for 60 min at 25°C. Fluorescence measurements were acquired at 620 and 665 nm using an Artemis plate reader (Cisbio, Bedford, MA), and FRET data were calculated as the 665/620 ratio. The signal values of WT SSTR2 were normalized from 0% to 100%. The signal values of mutant SSTR2 were indicated relative to those of WT SSTR2. The expression levels of SSTR2 constructs were monitored by Western blot analysis using FLAG antibody (Sigma A8592, St. Louis, MO, USA) and actin antibody (Sigma A5441, St. Louis, MO, USA) (*Figure 2—figure supplement 1*). The HEK293 cells were obtained from Korean Cell Line Bank (https://cellbank.snu.ac.kr). The characteristics of the cell line are as follows: origin-kidney, species-human, cellular morphology-epithelia, growth pattern-monolayer, histopathology-transformed primary embryonial kidney. The detailed STR data can be found at Korean Cell Line Bank (https://cellbank.snu.ac.kr) with KCBL No. 21,573. The cell line was tested negative for mycoplasma contamination.

## Acknowledgements

We thank the members of the Song and Lee Laboratories for technical assistance and helpful discussion. We also thank Dr Seung-Hee Lee for critical reading of this manuscript. We thank the staff of the cryo-EM facility at RIKEN Center for Biosystems Dynamics Research, Yokohama, Japan, and Korea Basic Science Institute (KBSI), Daejeon, Korea, for the data collection. This work was supported by a grant (NRF-2020M3A9G7103934 to JS and WL) from the National Research Foundation (NRF) of Korea. The software programs used for the processing were supported by SBGrid (https://www.sbgrid.org/). The computing resource was supported by the Global Science Experimental Data Hub Center (GSDC), Korea Institute of Science and Technology Information (KISTI), and by the data analysis hub, Olaf in the Institute of Basic Sciences (IBS) Research Solution Center.

## Additional information

### Competing interests

Ji-Hye Yun: is an employee at PCG-Biotech and holds a research director position. Ji-Joon Song, Weontae Lee: is a co-founder of PCG-Biotech. The other authors declare that no competing interests exist.

### Funding

| Funder | Grant reference number | Author |
| --- | --- | --- |
| National Research Foundation of Korea | NRF-2020M3A9G7103934 | Ji-Joon Song Weontae Lee |

The funders had no role in study design, data collection and interpretation, or the decision to submit the work for publication.

### Author contributions

Yunseok Heo, Eojin Yoon, Conceptualization, Data curation, Investigation, Visualization, Writing – original draft, Writing – review and editing; Ye-Eun Jeon, Ji-Hye Yun, Data curation, Investigation; Naito Ishimoto, Data curation; Hyeonuk Woo, Validation; Sam-Yong Park, Data curation, Supervision; Ji-Joon Song, Conceptualization, Data curation, Funding acquisition, Project administration, Supervision, Visualization, Writing – original draft, Writing – review and editing; Weontae Lee, Conceptualization, Funding acquisition, Project administration, Supervision, Writing – review and editing

### Author ORCIDs

Sam-Yong Park http://orcid.org/0000-0001-6164-8896
Ji-Joon Song http://orcid.org/0000-0001-7120-6311

**Decision letter and Author response**
Decision letter https://doi.org/10.7554/eLife.76823.sa1
Author response https://doi.org/10.7554/eLife.76823.sa2

## Additional files

### Supplementary files
• Transparent reporting form

### Data availability
The cryo-EM map and the model are to be deposited at EMDB (https://www.ebi.ac.uk/https://www.ebi.ac.uk/) and RCSB (https://www.rcsb.org/) data base with the accession codes of EMD-32543 and 7WJ5, respectively.

The following datasets were generated:

| Author(s) | Year | Dataset title | Dataset URL | Database and Identifier |
|---|---|---|---|---|
| Heo Y, Yoon E, Jeon YE, Yun JH, Ishimoto N , Woo H, Park SY, Song JJ, Lee W | 2022 | Cryo-EM structure of the human somatostatin receptor 2 complex with its agonist somatostatin delineates the ligand binding specificity | https://www.rcsb.org/structure/7WJ5 | RCSB Protein Data Bank, 7WJ5 |
| Heo Y, Yoon E, Jeon YE, Yun JH, Ishimoto N, Woo H, Park SY, Song JJ, Lee W | 2022 | Cryo-EM structure of the human somatostatin receptor 2 complex with its agonist somatostatin delineates the ligand binding specificity | https://www.ebi.ac.uk/emdb/EMD-32543 | EMDB, EMD-32543 |

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
