## [Editor Report]

This manuscript reports the cryoEM structure of somatostatin receptor 2 (SSTR2) bound to its agonist SST– 14 and a heterotrimeric G protein. In addition to presenting the structure itself, the authors include discussion and analysis of ligand recognition and subtype specificity, guided by AlphaFold2 modeling of other somatostatin receptor subtypes. Site– directed mutagenesis and signaling assay data attest to the importance of receptor– ligand contacts that contribute to subtype specificity. Somatostatin signaling is important in endocrine biology, including in diseases such as acromegaly and some cancers, and structures of somatostatin receptors will help illuminate the molecular details of somatostatin receptor signal transduction. Of note, the structure of SSTR2 was also separately reported by Robertson et al., Nat. Struct. Mol. Biol. 2022 while this manuscript was under review.

---

## [Decision Letter]

**Decision letter after peer review:**

Thank you for submitting your article "Cryo-EM structure of human somatostatin receptor 2 complex with its agonist somatostatin delineates the ligand binding specificity" for consideration by *eLife*. Your article has been reviewed by 2 peer reviewers, one of whom is a member of our Board of Reviewing Editors, and the evaluation has been overseen by Volker Dötsch as the Senior Editor. The reviewers have opted to remain anonymous.

Essential revisions:

1) The addition of a modest amount of functional data (i.e., binding or signaling assays) to test the importance of proposed receptor– ligand contacts is critical to understand which features directly contribute to subtype specificity. This could also broaden the appeal of the manuscript to other researchers with interests in GPCR signaling. Appropriate controls should be included for any receptor mutants tested.

2) Both reviewers noted that discussion of somatostatin receptor biology is brief and vague in several points. Please revise to include more precise and explicit discussion of disease relevance and of the role of SSTR signaling in normal biology.

*Reviewer #1 (Recommendations for the authors):*

The discussion of somatostatin biology is quite thin, and often rather vague. In the first sentence of the introduction, the authors state that SST is "implicated in several diseases". Much more precision would be helpful here. What does SST do in normal biology? Which diseases arise from dysfunction of SST signaling, and how? Some explicit discussion of the key unresolved structural and pharmacological questions would be helpful as well, to provide a clear rationale for the work described.

As noted above, some specific hypotheses around ligand recognition could be tested experimentally by mutagenesis or peptide engineering. Discussions based purely on structure (or alphafold predictions) are inherently speculative rather than definitive.

Ramachandran outliers should be fixed.

*Reviewer #2 (Recommendations for the authors):*

I would recommend the authors to introduce and discuss the literature regarding the pharmacology of somatostatin (SST– 14 in particular). For example, SST– 14 binds to all 5 receptors with similar affinities so it is hard to explain with just one structure how the residues that are not conserved in the binding pocket do regulate the binding specificity.

The authors also do not discuss at all the active state features of SSTR2 and only briefly the coupling with the Gi protein. Also, you describe the binding of a cyclic peptide. How does it compare to other known structures, if any? Could we learn something about cyclic peptide binding mode for GPCR?

[Editors' note: further revisions were suggested prior to acceptance, as described below.]

Thank you for resubmitting your work entitled "Cryo– EM structure of the human somatostatin receptor 2 complex with its agonist somatostatin delineates the ligand binding specificity" for further consideration by *eLife*. Your revised article has been evaluated by Volker Dötsch (Senior Editor) and a Reviewing Editor.

The manuscript has been improved but there are some remaining issues that need to be addressed, as outlined below:

1. Expression level data are necessary to evaluate the effects of point mutations presented in Figure 2D, as discussed in the previous reviews. It is possible that the attenuated signaling of the mutants reflects differences in expression level, rather than differences in ligand recognition.

2. Some additional discussion of the recent Robertson et al. paper would be helpful, highlighting where conclusions in the present manuscript confirm or extend data in the other paper.

---

## [Author Response]

Essential revisions:1) The addition of a modest amount of functional data (i.e., binding or signaling assays) to test the importance of proposed receptor– ligand contacts is critical to understand which features directly contribute to subtype specificity. This could also broaden the appeal of the manuscript to other researchers with interests in GPCR signaling. Appropriate controls should be included for any receptor mutants tested.

In the revised manuscript, we mutagenized the crucial residues involved in SST-14 binding and measured their functionality monitoring the inhibition of cAMP production upon forskolin stimulation in HEK-293 cells, which included as Figure 2D. This functional data validated the binding mode of SST-14 to SSTR2 observed in the cryo-EM structure. Regarding the ligand specificity among the isoforms, we realized that it is very complex to clearly explain the isoform-specificity based on our cryo-EM structure. The recently published SSTR2 structure (Robertson et al., 2022) during this revision shows that ECL2 is also involved in conferring the ligand specificity, and the our cryo-EM map near ECL2 is not atomic resolution preventing our further investigation. Therefore, we included the recently published data and toned down our discussion regarding the isoform specificity in the discussion.

2) Both reviewers noted that discussion of somatostatin receptor biology is brief and vague in several points. Please revise to include more precise and explicit discussion of disease relevance and of the role of SSTR signaling in normal biology.

In the revised manuscript, we added our discussion regarding somatostatin biology and the disease relevance for clarity.

Reviewer #1 (Recommendations for the authors):The discussion of somatostatin biology is quite thin, and often rather vague. In the first sentence of the introduction, the authors state that SST is "implicated in several diseases". Much more precision would be helpful here. What does SST do in normal biology? Which diseases arise from dysfunction of SST signaling, and how? Some explicit discussion of the key unresolved structural and pharmacological questions would be helpful as well, to provide a clear rationale for the work described.

We described the normal function of SST and discussed how the SST-SSTR axis can be targeted for disease treatment. In addition, we included several sentences to provide a clear rationale for the work.

As noted above, some specific hypotheses around ligand recognition could be tested experimentally by mutagenesis or peptide engineering. Discussions based purely on structure (or alphafold predictions) are inherently speculative rather than definitive.

In the revised manuscript, we mutagenized the crucial residues involved in SST-14 binding and function. We then measured their functionality monitoring the inhibition of cAMP production upon forskolin stimulation in HEK-293 cells, which included as Figure 2D. This functional data validated the binding mode of SST-14 to SSTR2 observed in the cryo-EM structure. Regarding the ligand specificity among the isoforms, we realized that it is very complex to clearly explain the isoform-specificity based on our cryo-EM structure. The recently published SSTR2 structure (Robertson et al., 2022) during this revision shows that ECL2 is also involved in conferring the ligand specificity, and the cryo-EM map near ECL2 is not atomic resolution preventing our further investigation. Therefore, we included the recently published data and toned down our discussion regarding the isoform specificity in the discussion.

Ramachandran outliers should be fixed.

We further refined the structure and there is no Ramachandran outlier in the revised manuscript.

Reviewer #2 (Recommendations for the authors):I would recommend the authors to introduce and discuss the literature regarding the pharmacology of somatostatin (SST– 14 in particular). For example, SST– 14 binds to all 5 receptors with similar affinities so it is hard to explain with just one structure how the residues that are not conserved in the binding pocket do regulate the binding specificity.

We introduced a few sentences regarding the pharmacology of SST. Regarding the isoform specificity, we agree with the reviewer in that this structure cannot sufficiently explain the isoform specificity. In addition, the recent published work by Skinoitis group during this revision shows that ECL2 plays a critical role in ligand specificity. Therefore, we referred to the recent work and toned down our discussion regarding the isoform specificity.

The authors also do not discuss at all the active state features of SSTR2 and only briefly the coupling with the Gi protein. Also, you describe the binding of a cyclic peptide. How does it compare to other known structures, if any? Could we learn something about cyclic peptide binding mode for GPCR?

In the revised manuscript, we compared our active SSTR2 with other GPCR structures and discussed the characteristics of the active conformation in the main text. In this regard, we included Figure1—figure supplement 4. In addition, we compared the cyclic peptide ligands among SSTR2 and MC4R, and discussed the features of the cyclic peptides in the results and added Figure1—figure supplement 5.

[Editors' note: further revisions were suggested prior to acceptance, as described below.]

The manuscript has been improved but there are some remaining issues that need to be addressed, as outlined below:1. Expression level data are necessary to evaluate the effects of point mutations presented in Figure 2D, as discussed in the previous reviews. It is possible that the attenuated signaling of the mutants reflects differences in expression level, rather than differences in ligand recognition.

In the revised manuscript, we included Western blot analysis to monitor the expression levels of SSTR2 constructs in HEK293 cells. We included these data as Figure 2—figure supplement 1 and Figure2—figure supplement 1-source data 1.

2. Some additional discussion of the recent Robertson et al. paper would be helpful, highlighting where conclusions in the present manuscript confirm or extend data in the other paper.

In the previous revision, we had included several sentences in page 6. “A recent report on the SSTR2 structure showed that extracellular loop 2 (ECL2) is involved in ligand-specific binding of SSTR2, which has highly variable sequences among the SSTR isoforms (Figure 3B)^21^. Therefore, several regions of SSTR2 including residues in the ECL2 and the inside of the binding pockets likely contribute to the ligand binding specificity. Together with the recent work by the Skiniotis group, our structural and sequence analysis revealed subtle differences in the ligand binding pocket in each SSTR isoform, providing a critical information to understand how each SSTR isoform specifically recognizes its cognate ligand and drug.” In this revision, we further discussed our findings compared with the other paper.